# CoDe: Semantic Color Reasoning and High-Fidelity Detail Synthesis for Underwater Image Enhancement

## Abstract

Underwater images often suffer from severe color distortion and texture degradation due to light absorption and scattering, posing huge challenges for visual perception and restoration. Recent diffusion-based underwater image enhancement (UIE) methods have shown remarkable performance, but most rely on customized architectures trained from scratch or lack auxiliary guidance beyond image-level inputs, which limit the model generalization and controllability. In this work, we propose a semantic **Co**lor reasoning and high-fidelity **De**tail synthesis UIE framework (CoDe), which fully leverages the synergy of diffusion models and vision-language models. It explicitly disentangles color and texture of underwater images: a fine-tuned LLaVA provides domain-invariant semantic color cues for robust color correction, while an SDXL-based generator restores high-frequency details for sharp reconstruction. Furthermore, we design an adaptive degradation-aware feature modulation module that fuses underwater and clean-domain representations, effectively suppressing noise interference during the denoising diffusion process. Extensive experiments on multiple underwater benchmarks demonstrate that CoDe achieves superior performance, significantly improving both color fidelity and texture preservation.

## 1 Introduction

Underwater images generally suffer from composite degradation caused by wavelength-dependent light absorption and scattering, resulting in color distortions and texture loss Jaffe (2002); Akkaynak & Treibitz (2019). These degradations not only hinder human visual perception but also seriously impair the reliability of downstream applications such as underwater robotics Lei et al. (2025), marine monitoring Jeong et al. (2024), and archaeological exploration Calantropio & Chiabrando (2024). Therefore, underwater image enhancement (UIE) has emerged as a fundamental yet challenging problem in computer vision and marine robotics.

Although numerous approaches have been proposed for UIE, they remain limited in the generalization and controllability. Physics-based models Hitam et al. (2013); Galdran et al. (2015); Xiang et al. (2018) rely on handcrafted assumptions or physical priors about underwater light propagation, but often fail in complex real-world conditions. Deep learning methods (CNNs/Transformers) Li et al. (2020); Peng et al. (2023) learn direct mappings but struggle to handle diverse degradations and strong domain shifts. Recent advances in diffusion-based methods Ho et al. (2020); Rombach et al. (2022) have boosted restoration quality. However, most approaches either require customized architectures trained from scratch, resulting in enormous computational costs Zhao et al. (2024); Song et al. (2025), or rely solely on degraded image inputs Xia et al. (2025); Ou et al. (2025), which constrains their robustness under domain shifts and weakens controllability in practical applications.

In fact, the central challenge in UIE is that color correction and texture restoration are fundamentally different Awan & Mahmood (2024); Rani et al. (2025). Color distortions arise from wavelength-dependent absorption, which introduces irreversible information loss that cannot be resolved at the pixel level. Unlike texture degradation, which retains structural cues and can be recovered through fine-grained image-level modeling, accurate color correction requires semantic-level reasoning to infer the plausible true colors of objects. On the other hand, texture degradation mainly stems from

scattering and demands detail-oriented reconstruction. Therefore, it is feasible to disentangle the two parts for effective restoration.

However, existing decomposition methods Zhou et al. (2025); Xue et al. (2025) generally couple color correction and texture recovery within a single generative process, relying solely on image-level information. This entanglement often results in inherent trade-offs: enhanced color fidelity at the cost of blurred details, or sharper structures accompanied by distorted colors. Inspired by modernized text-to-image models such as Stable Diffusion (SD) Kim et al. (2025); Yang et al. (2025), which can accurately synthesize high-resolution and semantically coherent images, we argue that decoupling color and texture with multi-modal guidance provides a principled path for UIE.

To address these challenges, we propose CoDe, a multi-modal decoupling framework that explicitly separates the modeling of semantic color and high-fidelity texture. A fine-tuned LLaVA model extracts domain-invariant textual color caption from underwater images, enabling robust and semantic-level color correction. A diffusion-based generator (SDXL) reconstructs high-frequency details and restores sharp textures. Meanwhile, a lightweight feature modulation module integrates underwater and noised latent feature representations, which effectively suppresses noise interference during iterative denoising and improves structural preservation. Extensive experiments on multiple underwater benchmarks demonstrate that CoDe achieves superior color fidelity and texture restoration, outperforming state-of-the-art methods in both quantitative metrics and perceptual quality.

In summary, our contributions are four-fold: **1)** We propose CoDe, a multi-modal UIE framework that enhances underwater images at both the text and image levels. **2)** We decouple color correction and detail restoration by combining LLaVA-based color reasoning with SDXL-based generative detail recovery. **3)** We design an adaptive degradation-aware modulation module that selectively fuses underwater texture features with diffusion latents, achieving enhanced fidelity and sharpness. **4)** Extensive experiments on multiple real-world and synthetic underwater datasets demonstrate that our method achieves state-of-the-art performance, consistently improving both color fidelity and texture preservation compared with existing UIE methods.

## 2 RELATED WORK

**Underwater Image Enhancement.** Underwater image enhancement (UIE) has been widely studied as a fundamental direction to improve the visual quality and usability of underwater imagery. Traditional UIE methods rely on hand-crafted prior or simplified physical models, such as CLAHE Hitam et al. (2013), white-balance Sanila et al. (2019), gamma correction Xiang et al. (2018), underwater dark-channel prior Drews et al. (2016), red-channel deficiency prior Galdran et al. (2015) and super-laplacian reflectance prior Peng & Cosman (2017). These methods are adopted for the negligible computational cost, yet they frequently over-enhance images and ignore the wavelength-dependent attenuation inherent to underwater scenes, leading to failure in real conditions.

With the rise of deep learning, data-driven UIE methods have become popular. CNN-based methods leverage convolutional backbones with physics-inspired priors, e.g., UWCNN Li et al. (2020) adopts a residual CNN with physics-based loss, SC-Net Fu et al. (2022) introduces spatial- and channel-wise normalization to adapt to diverse water types. Transformer-based methods further enhance long-range modeling, such as UFormer Peng et al. (2023), which incorporates channel-spatial attention and multi-color space loss to correct color casts. GAN-based methods have been explored: Semi-UIR Huang et al. (2023) employs a mean-teacher framework with contrastive loss, while PUGAN Cong et al. (2023) integrates physics priors with adversarial training. There are also several methods with distanglement learning Liu et al. (2024b). FiveAPLus Jiang et al. (2023) adopts a divide-and-conquer strategy for real-time enhancement by separately addressing color and detail. WPFNet Liu et al. (2024a) leverages wavelet-domain decomposition to capture multi-scale frequency features. Despite these advances, many methods still generate results that are perceptually unsatisfactory, with either residual color distortion or loss of fine-grained texture details.

**Diffusion-based Underwater Image Enhancement.** Diffusion models have recently been explored for UIE due to the strong generative capability, which offers a novel solution to simultaneously address color correction and detail recovery. Early works adapt customized diffusion architectures to underwater domain. WF-Diff Zhao et al. (2024) introduces a wavelet–Fourier conditional diffusion framework to explicitly model frequency-dependent attenuation. DM_underwater Tang et al. (2023)

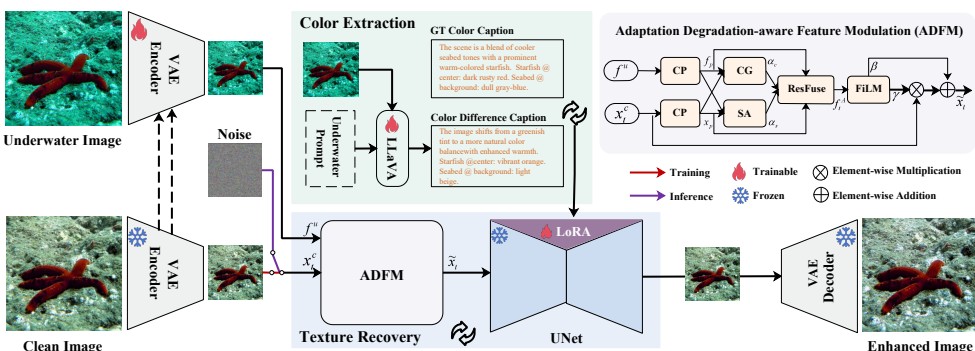

Figure 1: The proposed multi-modal decouping framework for underwater image enhancement.

employs a lightweight transformer-based denoising network with a non-uniform timestep schedule to accelerate sampling. CPDM Shi & Wang (2024) compensates low-level features in each diffusion step, improving robustness under diverse conditions. UW-DiffPhys Bach et al. (2024) combines physics-based priors with a simplified denoising diffusion framework, while DiffColor Chang et al. (2025) jointly applies global color correction and cross-spectral refinement to recover color and texture. Other works incorporate cross-domain or perceptual guidance. CLIP-UIE Liu et al. (2025) leverages CLIP-based classification to inject natural image priors into UIE, while UIEDP Du et al. (2025) demonstrates that Stable Diffusion can be fine-tuned on only 300 labeled underwater samples. UIE_CLIP Cao et al. (2025) integrates CLIP-based perceptual losses and curriculum contrastive regularization to align outputs with human visual preferences. More recent efforts propose degradation-aware conditioning, e.g., DACA-Net Huang et al. (2025), or compress diffusion into fewer steps, as in SSL-Diff Wu et al. (2025). Overall, diffusion-based UIE has achieved notable improvements in perceptual quality. However, most existing methods remain focused on low-level cues, while semantic-level guidance (e.g., language-driven color reasoning) and explicit disentanglement of color and texture have been underexplored. These gaps motivate our proposal.

## 3 PROPOSED METHOD

### 3.1 OVERVIEW

As illustrated in Fig. 1, we propose a multi-modal decoupling framework for underwater image enhancement (UIE), dubbed CoDe, which explicitly disentangles color and texture modeling. Color cues are extracted as textual captions via a vision–language model (e.g., LLaVA Liu et al. (2023)), providing domain-invariant semantic information that remains robust under pixel-level degradations. Texture and structural details are subsequently reconstructed with a diffusion backbone (e.g., SDXL), which excels at high-fidelity generation. Instead of directly injecting degraded underwater features as the condition at each denoising step, it risks noise accumulation and artifact generation. We also introduce a lightweight degradation-aware feature modulation module to further stabilize the process. It smoothly reconciles the underwater and noised features, enabling effective guidance for color correction and contrast adjustment while suppressing degradation propagation.

### 3.2 VLM-BASED DOMAIN-INVARIANT COLOR EXTRACTION

Color cast caused by wavelength-dependent absorption represents a fundamental degradation in underwater imaging Wen et al. (2025). To explicitly disentangle color correction from texture restoration, we expect to capture ideal semantic color cues under normal conditions into the generative process. Although paired underwater–clean images are available during training, clean references are absent at inference. To bridge this gap, we propose to extract domain-invariant color representations through visual-language reasoning. Specifically, we fine-tune LLaVA-1.5-13b to translate pixel-level underwater color degradations into semantic color captions that guide subsequent diffusion-based restoration. The procedure includes training corpus construction, fine-tuning via visual instruction, and inference for semantic guidance.

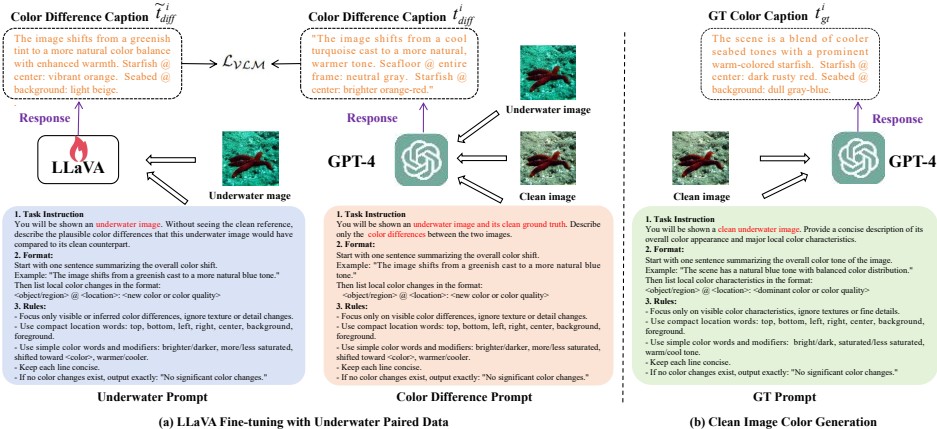

Figure 2: Illustration for LLaVA fine-tuning and color caption generation.

**Training Corpus Construction.** To adapt LLaVA in underwater domain, we need to prepare the training corpus first. As illustrated in Fig. 2, to obtain a reliable caption supervised information, we employ a large language model (GPT-4) to capture color difference captions $t^i_{diff}$ between each pair $(I^i_u, I^i_c)$, where $I_u$ and $I_c$ denote underwater and clean images, respectively. The caption mainly includes global color tones and local object-level color properties. These captions capture the semantic shift from degraded to clean colors in a compact textual form.

**Fine-tuning via Visual Instruction.** We then fine-tune LLaVA using LoRA-based adaptation to adapt its visual reasoning capability to underwater domains. Only underwater images $I^i_u$ are used as input, and $t^i_{diff}$ is used as supervision. Formally, the objective is:

$$\mathcal{L}_{\mathcal{VLM}} = \sum_{i=1}^{N} \mathcal{L}_{\mathcal{CE}}(\hat{t}^i_{diff}, t^i_{diff}), \tag{1}$$

where $\hat{t}^i_{diff}$ is the predicted caption from $I^i_u$ and $\mathcal{L}_{\mathcal{CE}}$ denotes the cross-entropy over token sequences. This training strategy encourages LLaVA to abstract color semantics that approximate clean-domain distributions, despite only observing degraded inputs. It effectively learns to acquire color semantics invariant to degradation patterns.

**Inference for Semantic Guidance.** During inference, given a degraded image $I^i_u$, the fine-tuned LLaVA generates $\hat{t}_{diff}$, which describes how its color should shift toward the clean domain. The textual embedding of $\hat{t}_{diff}$ is then injected into the text encoder of SDXL, guiding the diffusion process toward clean and natural color restoration. This design achieves a principled decoupling: color correction is governed by semantic-level textual guidance through robust visual-language reasoning, while texture restoration is handled by the generative backbone. The separation ensures that color-related degradations are addressed in a compact and domain-invariant representation space, preventing pixel-level artifacts from propagating through the enhancement pipeline.

## 3.3 DIFFUSION-BASED HIGH-FIDELITY TEXTURE RECOVERY

After obtaining the color textual caption from LLaVA, we employ it as a semantic conditioning signal for the SDXL model. Specifically, the caption is encoded via SDXL's text encoder and incorporated into the cross-attention layers to guide the generative process. This conditioning strategy enables SDXL to reconstruct high-frequency textures and structural details while remaining consistent with the color semantics inferred by the vision-language model. By decoupling color and texture property, we alleviate the burden on the diffusion model to simultaneously correct color cast and recover fine details, thereby improving both the perceptual quality and fidelity of the output.

**Adaptive Degradation-aware Feature Modulation (ADFM).** To guide the ideal content, a naive way to condition the diffusion model on underwater features $f^u$ is to directly combine them with the noisy latent $x^c_t$ at each denoising step. However, it is problematic: as the diffusion process pro-

gresses, $x_t^c$ gradually becomes refined, whereas $f^u$ remains fixed and still encodes domain-specific degradations. Simply injecting $f^u$ thus risks propagating underwater artifacts into the generation process. To mitigate this, we propose a lightweight and Adaptive Degradation-aware Feature Modulation module that adaptively fuses the features from the degraded image with the evolving diffusion features. As shown in Fig. 1, it operates at each step and contains the following components:

*Channel projection (CP).* Features from both the underwater image and current noisy latent are projected into a common channel dimension via $1 \times 1$ convolution with weights $W_f$ and $W_x$:

$$f_p = W_f f^u, \quad x_p = W_x x_t^c. \tag{2}$$

*Channel-wise gating (CG).* A gating vector is obtained by global average pooling and a lightweight multi-layer perceptron (MLP), effectively controlling the inter-channel influence across features:

$$\alpha_c = \sigma\big(\text{MLP}\big(\text{Concat}(\text{GAP}(f_p), \text{GAP}(x_p))\big)\big), \tag{3}$$

where $\sigma$ is the sigmoid function. Concat is the concatenation operation.

*Spatial attention (SA).* The spatial attention module (Conv-ReLU-Conv-Sigmoid) computes a soft spatial mask $\alpha_s \in [0, 1]^{H \times W}$ to further regulate feature fusion in a position-aware manner:

$$\alpha_s = SA\big(\text{Concat}(f_p, x_p)\big). \tag{4}$$

*Residual-style fusion (ResFuse).* The projected features are combined in a residual manner, modulated by the channel and spatial attention weights to obtain the fused features $f_t^A$:

$$f_t^A = x_p + \alpha_c \odot (f_p - x_p), \quad f_t^A \leftarrow (1 - \alpha_s) \odot x_p + \alpha_s \odot f_t^A. \tag{5}$$

*FiLM-based modulation (FiLM).* Finally, the fused feature $f_t^A$ is used to compute affine parameters for a FiLM-based modulation Perez et al. (2018) of the original noisy latent $x_t^c$:

$$\tilde{x}_t = \gamma(f_t^A) \odot x_t^c + \beta(f_t^A), \tag{6}$$

where $\gamma$ and $\beta$ are implemented by $1 \times 1$ convolution and initialized as 1 and 0 for stable training.

This design offers several advantages: (*1*) it is computationally efficient, relying only on lightweight projections and attention mechanisms; (*2*) the gating mechanism adaptively reduces the influence of degraded features as denoising proceeds, minimizing artifact propagation; and (*3*) the FiLM modulation enables underwater-specific adjustments in color and contrast without overriding clean-domain structures. Together, ADFM allows the diffusion model to leverage underwater priors while avoiding the artifacts introduced by naive feature injection.

**Parameter-efficient Fine-tuning with LoRA.** Given the substantial computational footprint of SDXL, we employ Low-Rank Adaptation (LoRA) Hu et al. (2022) to fine-tune the model. LoRA freezes the original pre-trained weights and introduces trainable low-rank decompositions into the attention layers. This way significantly reduces the number of trainable parameters and GPU memory requirements, while still enabling effective adaptation to the underwater domain. By combining ADFM and LoRA, our method achieves a favorable balance between performance and efficiency.

### 3.4 OPTIMIZATION

To train the proposed CoDe, enabling it to accurately decompose the degradation process in underwater images and generate high-quality clear outputs, we adopt a multi-component loss function. It incorporates not only the core loss driving the diffusion model's denoising process but also auxiliary losses operating in the image space to specifically address structural preservation and perceptual realism crucial for underwater imagery.

**Noise Prediction Loss.** Given a clean latent representation $z_0$ and a randomly sampled timestep $t$, Gaussian noise $\epsilon \sim \mathcal{N}(0, I)$ is injected to obtain $z_t$, and the denoising U-Net predicts the added noise $\hat{\epsilon}_\theta(z_t, t, c, I_u)$. The basic diffusion loss is formulated as:

$$\mathcal{L}_{\text{noise}} = \mathbb{E}_{z_0, \epsilon, t} \left[\|\epsilon - \hat{\epsilon}_\theta(z_t, t, c, I_u)\|_1\right], \tag{7}$$

where $c$ denotes the color-aware caption extracted from the fine-tuned LLaVA ($\hat{t}_{diff}$) model and pre-trained GPT-4 ($t_{gt}$). And these two captions are alternately input.

**Fidelity Loss.** To ensure structural preservation and pixel-level faithfulness, we impose an additional reconstruction constraint in the image space:

$$\mathcal{L}_{\text{fidelity}} = \|I_{pred} - I_c\|_1, \tag{8}$$

where $I_{pred}$ and $I_c$ denote the reconstructed and ground-truth clean images, respectively.

**Color Consistency Loss.** Since underwater images often suffer from wavelength-dependent attenuation, we introduce a channel-level color constraint to regularize color distributions:

$$\mathcal{L}_{\text{color}} = \sum_{c \in \{R,G,B\}} \left( \mu_c(I_{pred}) - \mu_c(I_c) \right)^2, \tag{9}$$

where $\mu_c(\cdot)$ denotes the average intensity of channel $c$.

The overall training objective integrates the above components as:

$$\mathcal{L} = \mathcal{L}_{\text{noise}} + \lambda_1 \mathcal{L}_{\text{fidelity}} + \lambda_2 \mathcal{L}_{\text{color}}, \tag{10}$$

where $\lambda_1$ and $\lambda_2$ balance the contributions of fidelity and color constraints. And we empirically set as $\lambda_1 = 0.1$ and $\lambda_2 = 0.05$. This composite loss ensures that the model not only learns robust denoising but also produces visually faithful and color-consistent outputs.

**Training Strategy.** To reduce the distribution gap between training and inference we adopt a hybird caption strategy. During SDXL fine-tuning, each underwater image is paired with both a clean-domain caption (from the ground truth) and a predicted caption (generated by LLaVA). We gradually shift from clean captions to LLaVA captions over the training process, enabling the diffusion model to learn from accurate supervision while remaining robust to imperfect text conditions in inference.

## 4 EXPERIMENTS

### 4.1 EXPERIMENTAL SETTING

**Datasets.** Following previous studies Du et al. (2025); Fan et al. (2026), we conducted a comprehensive comparison with state-of-the-art methods on four widely used UIE benchmarks: UIEB Li et al. (2019a), EUVP Islam et al. (2020), C60 Li et al. (2019a) and U45 Li et al. (2019b) datasets. The UIEB dataset contains 890 paired underwater and clean images. We adopt 800 pairs for training and the remaining 90 pairs for testing. The EUVP dataset provides $11,435$ paired images for training and we select 200 pairs for testing. In addition, UIEB includes a set of 60 challenging underwater images (C60) without reference ground truth. The U45 dataset comprises 45 carefully selected real-world underwater images, serving as a widely used benchmark for no-reference evaluation. For both C60 and U45, we directly employ the model weights trained on UIEB for testing.

**Implementation Details.** We adopt SDXL Podell et al. (2024) as the backbone diffusion model, and reuse its VAE encoder as the underwater image feature encoder. The model is optimized using Adam with $\beta_1 = 0.9$, $\beta_2 = 0.999$, and $\epsilon = 10^{-8}$. All images are resized to $256 \times 256$ for training and testing, and upsampled to $512 \times 512$ to better match the training resolution of SDXL. For the training process, we adopt a two-stage fine-tuning strategy. We first pre-train the proposed feature modulation module for $3,000$ iterations with an initial learning rate of $5 \times 10^{-5}$ and no text inputs. We then fine-tune the entire network including the underwater encoder, the modulation module, and LoRA parameters in an end-to-end manner for $40,000$ iterations. During fine-tuning, the learning rate for the underwater encoder and other components are initialized as $5 \times 10^{-6}$ and $1 \times 10^{-5}$, respectively. Tbe LoRA parameters are $r = 32$ and $\alpha = 64$. The learning rates are updated using the cosine annealing schedule Loshchilov & Hutter (2017). For inference, we adopt Euler scheduler Karras et al. (2022) with 20 sampling steps.

**Evaluation Metrics.** In the above test sets, each underwater image in UIEB and EUVP has a corresponding ground-truth reference, whereas C60 and U45 contain only underwater images. Accordingly, we evaluate UIE performance with both full-reference and no-reference image quality metrics. For full-reference evaluation, we employ PSNR and SSIM, which measure pixel-level fidelity and structural similarity to the references. For no-reference assessment, we adopt two metrics tailored to UIE: UIQM Panetta et al. (2015), which accounts for contrast, sharpness, and colorfulness, and UCIQE Yang & Sowmya (2015), which considers brightness instead of sharpness.

### 4.2 COMPARISONS WITH THE STATE-OF-THE-ART

To validate the effectiveness of our method, we compare it against several state-of-the-art UIE approaches, spanning both conventional CNN or transformer based methods (e.g., UWCNN Li et al.

Table 1: Quantitative comparison on four underwater image enhancement benchmarks. Best results are highlighted in **bold**.

| Methods | UIEB | | EUVP | | C60 | | U45 | |
|---|---|---|---|---|---|---|---|---|
| | PSNR↑ | SSIM↑ | PSNR↑ | SSIM↑ | UIQM↑ | UCIQE↑ | UIQM↑ | UCIQE↑ |
| UWCNN Li et al. (2020) | 15.2818 | 0.7065 | 19.6235 | 0.7530 | 2.4616 | 0.3402 | 3.1350 | 0.3819 |
| SC-Net Fu et al. (2022) | 21.9168 | 0.8807 | 20.8981 | 0.8040 | 2.7360 | 0.3806 | 3.3473 | 0.4069 |
| UFormer Peng et al. (2023) | 21.5591 | 0.8155 | 24.8880 | 0.8052 | 2.8247 | 0.3743 | 3.3098 | 0.3832 |
| Semi-UIR Huang et al. (2023) | 20.2539 | 0.8186 | 19.3955 | 0.7244 | 2.8390 | 0.4309 | 3.3656 | 0.4274 |
| FiveAPlus Jiang et al. (2023) | 23.1982 | 0.9121 | 21.5655 | 0.7918 | 2.6651 | 0.4054 | 3.3939 | 0.4213 |
| PUGAN Cong et al. (2023) | 23.3641 | 0.8879 | 25.0827 | 0.8224 | 3.0559 | 0.4454 | 3.3755 | 0.4460 |
| WPFNet Liu et al. (2024a) | 22.2905 | 0.8943 | 22.6200 | 0.7801 | 3.0153 | 0.3756 | 3.4886 | 0.3897 |
| CLIP-UIE Liu et al. (2025) | 24.3167 | 0.9265 | 18.1056 | 0.7483 | 2.9091 | 0.4405 | 3.1607 | 0.4561 |
| UIEDP Du et al. (2025) | 20.2539 | 0.8186 | 21.8403 | 0.7925 | 3.0594 | 0.4341 | 3.4590 | 0.4386 |
| UIE_CLIP Cao et al. (2025) | 24.7390 | **0.9280** | 20.6598 | 0.7818 | 2.8521 | 0.4196 | 3.4493 | 0.4216 |
| **CoDe (Ours)** | **24.9572** | 0.9262 | **25.3359** | **0.8672** | **3.3552** | **0.4931** | **3.5345** | **0.5383** |

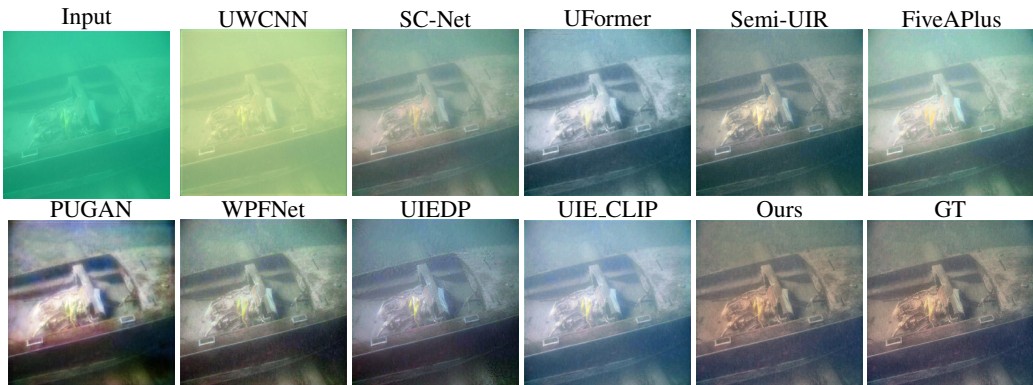

Figure 3: Visual comparisons of our CoDe with other state-of-the-art methods on UIEB dataset.

(2020), SC-Net Fu et al. (2022), UFormer Peng et al. (2023), Semi-UIR Huang et al. (2023), FiveAPLus Jiang et al. (2023), WPFNet Liu et al. (2024a)) and recent generative or diffusion-based frameworks (e.g., PUGAN Cong et al. (2023), CLIP-UIE Liu et al. (2025), UIEDP Du et al. (2025), UIE_CLIP Cao et al. (2025)).

**Quantitative Results.** As presented in Table 1, our method almost surpasses existing approaches across all datasets and evaluation metrics. On the paired UIEB benchmark, our framework achieves superior performance with a PSNR of 24.9572 dB and an SSIM of 0.9262, slightly outperforming the strongest baseline UIE_CLIP by 0.2182 dB in PSNR while maintaining comparable structural similarity. On EUVP, our method shows a clear advantage, reaching 25.3359 dB PSNR and 0.8672 SSIM, which exceeds the second-best PUGAN by 0.2532 dB and 0.0448 in SSIM. For the more challenging no-reference benchmarks, our approach significantly improves perceptual quality: on C60, UIQM and UCIQE are boosted to 3.3552 and 0.4931, respectively, marking the largest margins over prior methods. Similarly, on U45, our method yields 3.5345 UIQM and 0.5383 UCIQE, demonstrating its robustness in real-world scenarios. These consistent gains validate the effectiveness of our color–texture decoupling strategy and the proposed degradation-aware modulation module in enhancing both fidelity and perceptual quality.

**Qualitative Results.** As illustrated in Fig. 3, CNN-based approaches such as UWCNN and UFormer tend to leave residual color casts or produce over-smoothed results, leading to loss of fine details. Recent diffusion- and GAN-based methods (e.g., PUGAN, CLIP-UIE) achieve improved global color balance, yet often generate unnatural textures or fail under severe greenish and bluish degradations. In comparison, our method delivers more faithful restorations with vivid but realistic colors and sharper structures. In particular, fine details such as coral textures and object boundaries are better preserved, while color tones remain natural and consistent with human perception. These observations corroborate the quantitative improvements, highlighting the effectiveness of our underwater-

Table 2: Ablation studies of different components for UIE on UIEB dataset.

| Case | Model | PSNR↑ | SSIM↑ | UCIQE↑ | UIQM↑ |
|------|-------|-------|-------|--------|-------|
| 1 | Baseline | 17.5200 | 0.5783 | 0.3780 | 2.5141 |
| 2 | + Pre-trained LLaVA for overall caption | 18.1249 | 0.6043 | 0.3901 | 2.6417 |
| 3 | + Pre-trained LLaVA with PUGAN input | 18.3947 | 0.6681 | 0.4072 | 2.7952 |
| 4 | + Pre-trained LLaVA for color caption | 17.8179 | 0.6349 | 0.4384 | 2.6833 |
| 5 | + Fine-tuned LLaVA for color caption | 22.3916 | 0.7703 | 0.4461 | 2.8365 |
| 6 | + ADFM | 23.2580 | 0.8703 | 0.4461 | 3.3365 |
| 7 | Ours | 24.9572 | 0.9262 | 0.4937 | 3.4249 |

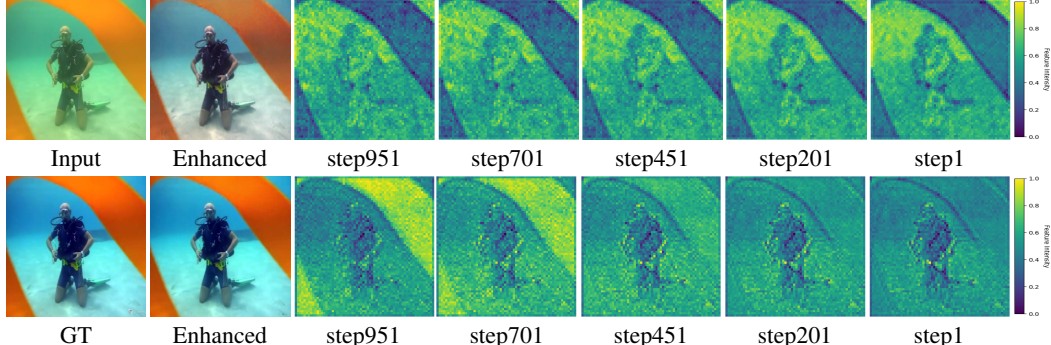

Figure 4: The visualized comparisons for the fused features without (top) and with (bottom) the degradation-aware feature modulation module.

aware alignment module and color–texture decoupling design in handling diverse underwater conditions. More visual results can be seen in the supplemental materials.

**Discussion.** These results verify that our approach not only advances the quantitative state-of-the-art but also achieves superior visual fidelity and perceptual realism. Compared with the previous diffusion-based methods such as UIEDP and CLIP-UIE, our framework avoids the common trade-off between color fidelity and texture sharpness by explicitly disentangling color and texture modeling. Moreover, the modulation module stabilizes cross-domain conditioning, preventing artifacts that often appear when degraded underwater features are directly injected into the diffusion process. Overall, our method demonstrates that integrating multi-modal guidance and lightweight alignment into diffusion models provides a principled and effective solution for underwater image enhancement. Beyond underwater imagery, the proposed framework can be readily extended to other cross-domain enhancement tasks such as low-light image enhancement and dehazing, highlighting its general applicability.

### 4.3 ABLATION STUDY

To evaluate the effectiveness of individual components, we conduct a comprehensive ablation study on UIEB dataset, examining the roles of color–texture decoupling, the degradation-aware modulation module, and the hybrid caption training strategy. The baseline only uses the underwater dataset to fine-tune SDXL with two-stage training. This analysis disentangles the effect of each design choice and demonstrates that jointly leveraging texture guidance, feature modulation, and tailored training strategy is essential for achieving robust underwater image enhancement.

**Effect of Textual Guidance.** To demonstrate the effect of text caption, we first generate the overall caption by pre-trained LLaVA with the prompt "Describe this image and its style in a very detailed manner" about the underwater and clear images in Case2. It is observed consistent improvements across all metrics (e.g., PSNR increases from 17.5200 to 18.1249, SSIM from 0.5783 to 0.6043), indicating that LLaVA-generated captions during training and inference provides extra semantic cues for color correction and structure preservation. Considering that there exists domain gap between underwater and clean images, their corresponding text caption is different. Therefore, we employ the pre-trained UIE model (PUGAN) to preprocess the underwater image during inference and re-generate the caption with pre-trained LLaVA to input to SDXL in Case3. It achieves even larger

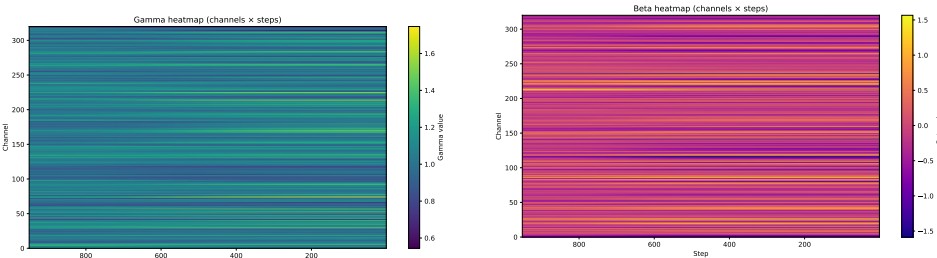

Figure 5: The visualized heatmap for $\gamma$ and $\beta$. Zoom in for better view.

gains (PSNR 18.3947, SSIM 0.6681), suggesting that pre-enhanced underwater images with caption generation yields more accurate and reliable textual descriptions, which in turn strengthen the guidance effect. These results highlight that high-quality textual supervision is crucial for enhancing the semantic consistency of color restoration and texture reconstruction in underwater scenarios.

**Effect of Color–Texture Decoupling.** To demonstrate the effect of color caption, we employs LLaVA-generated color-only descriptions during training and inference in Case4, leads to performance degradation compared with Case3 (e.g., PSNR drops from 18.3947 to 17.8179, SSIM from 0.6681 to 0.6349). This decline indicates that directly constraining the model with imperfect color descriptions introduces a mismatch between the textual guidance and the ground-truth distribution, thereby impairing the overall enhancement quality. To address this issue, Case5 fine-tunes LLaVA using GPT-4–generated captions that explicitly capture the color discrepancies between underwater and reference images. This adaptation allows the model to produce captions that more faithfully align with the ground-truth color distribution, resulting in a substantial performance boost (PSNR 22.3916, SSIM 0.7703). These results demonstrate that disentangling color from texture and providing precise, well-aligned color supervision is essential for robust semantic guidance in UIE.

**Effect of Adaptive Degradation-aware Feature Modulation.** Building upon the color-guided framework in Case5, Case6 further integrates ADFM to bridge the gap between underwater and clean image features. It effectively suppresses the noise of underwater degradation information in texture reconstruction, leading to a notable improvement across all metrics (PSNR increases from 22.39 to 23.2580, SSIM from 0.7703 to 0.8703). Besides, we also present the visualized analysis about the modulated features and the distributions of $\gamma$ and $\beta$ in Fig. 4 and Fig. 5. Our ADFM focuses its modulation on regions with severe color degradation, leading to better color fidelity in the enhanced outputs. The heatmaps of $\gamma$ and $\beta$ demonstrate that the modulation module provides stable yet channel-specific adjustments across denoising steps, enabling dynamic feature scaling and shifting that improves robustness and degradation removal.

**Effect of Hybrid Caption Strategy.** In Case7, we integrate the hybrid caption training strategy, which combines both color difference caption and the direct color caption guidance. Compared with Case6, this strategy further enhances the robustness of textual conditioning, especially in challenging cases with severe color distortions. Quantitatively, Case7 achieves the best overall performance, with PSNR improving from 23.2580 to 24.9572, SSIM from 0.8703 to 0.9262, and UIQM from 3.3365 to 3.4249. These gains demonstrate that hybrid captioning provides complementary information, which guides the correction of dominant casts to achieve superior perceptual quality.

## 5 CONCLUSION

In this paper, we propose a decoupled underwater image enhancement framework that integrates multi-modal understanding with diffusion-based generation. By fine-tuning LLaVA on paired underwater–clean data, our method extracts color semantics and injects them into SDXL as guiding prompts, effectively disentangling color correction from texture restoration. This design yields perceptually natural results, consistent quantitative gains, and parameter-efficient training via LoRA. While promising, the approach still relies on paired data quality, and diffusion inference remains slower than CNN- or GAN-based alternatives, limiting real-time deployment. Future work will focus on scaling to larger and more diverse datasets, developing lightweight diffusion variants for efficiency, and coupling enhancement with downstream tasks such as detection and segmentation.

## 6 ETHICS STATEMENT

This work focuses on improving image visibility in challenging aquatic environments. All datasets used in this study are publicly available benchmark datasets and do not involve private or sensitive personal data. Our method does not raise direct societal or ethical risks, as it is intended for benign applications such as marine monitoring, ecological research, and underwater robotics. We acknowledge that image enhancement techniques could potentially be misused in surveillance scenarios, but our design and experimental scope are limited to scientific and environmental use cases. We therefore believe that this work adheres to the ICLR Code of Ethics.

## 7 REPRODUCIBILITY STATEMENT

We make extensive efforts to ensure the reproducibility of our work. Detailed descriptions of the model architecture, training configuration, and hyper-parameters are provided in Sec. 3–4 of the main paper and in the appendix. The paired and unpaired underwater datasets used in our experiments are all publicly available. The implementation of our diffusion backbone and fine-tuned vision–language model follows publicly available frameworks. In addition, ablation studies and visual comparisons are provided to validate the stability and robustness of our framework.

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

## A   APPENDIX

In the appendix, we mainly give more details to further validate our proposed method. Specifically, we analyze different caption conditions and plot the visualization of the feature modulation parameters to show how they dynamically guide the diffusion process. In addition, we also give more qualitative results for both ablation study and comparison with state-of-the-art methods to show the subjective performance of our model in various scenarios.

### A.1   ANALYSIS OF DIFFERENT CAPTION CONDITIONS

Here, we mainly illustrate the effect of different caption sources on semantic guidance for SDXL in Fig. 6. When directly applying pre-trained LLaVA with underwater or enhanced (PUGAN Cong et al. (2023)) inputs, the generated captions usually contain redundant information or hallucinated content (e.g., "black and white style," "blurry blue background"), which provide weak or even misleading color cues. In contrast, captions from our fine-tuned LLaVA are more concise and more explicit about color difference (e.g., "coral richer orange-brown" or "water more blue"), and are closer to the clean-domain semantics. To build reliable supervision, we use GPT-4 to generate the ideal captions, which aims to describe consistent and domain invariant color shifts. Compared to pre-trained LLaVA with clean inputs, captions from GPT-4 are more accurate and less misleading towards irrelevant arts. Therefore, the results demonstrate that when using GPT-4 generated supervision to fine-tune LLaVA, we can get reliable semantic color alignment, which greatly improves the controllability and fidelity of diffusion based UIE.

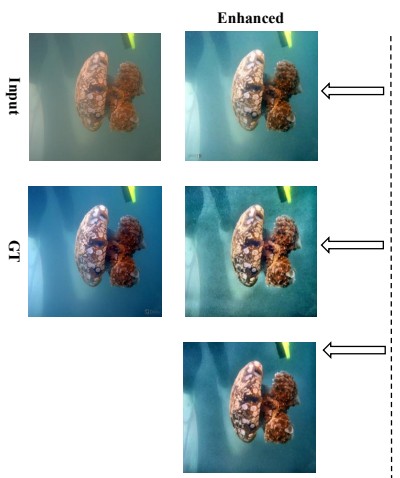

**Overall caption with pre-trained LLaVA for underwater input:**
The image features a close-up view of a large sea animal, possibly a jellyfish or a sea anemone, floating in the ocean. The animal is surrounded by water, and its body is covered in white spots. The creature appears to be resting on the ocean floor, with its body occupying a significant portion of the image. The scene is captured in a black and white style, which adds a sense of depth and contrast to the image. The focus on the sea animal and its intricate details creates a captivating and artistic representation of the underwater world.

**Overall caption with pre-trained LLaVA for enhanced input (PUGAN):**
The image features a close-up view of a sea creature, possibly a jellyfish or a sea anemone, sitting in the water. The creature is positioned in the center of the frame, with its body occupying a significant portion of the image. The creature's body is covered in a pattern of white spots, giving it a unique and intricate appearance. The background of the image is a blurry blue, which adds to the focus on the sea creature and creates a sense of depth in the scene. The overall style of the image is artistic and captures the beauty of the underwater world.

**Color difference caption with fine-tuned LLaVA for underwater input:**
The image shifts from a muted tone to a more vibrant and saturated appearance. Coral @ center: richer orange-brown. Water @ background: more blue. Sunlit area @ top-left: brighter white. Coral surface details @ center: enhanced contrast and color depth.

**Color difference caption with GPT-4 with for paired underwater and GT input:**
The image shifts from a muted, warm cast to a clearer and cooler tone. Jellyfish @ center: deeper brown with more contrast. Background water @ entire frame: less green, more blue. Reflections @ left: brighter white.

**Overall caption with Pre-trained LLaVA for GT input:**
The image features a close-up view of a large sea creature, possibly a jellyfish or a sea anemone, floating in the ocean. The creature is positioned in the center of the frame, with its body occupying a significant portion of the image. The creature's body is covered in numerous small white spots, giving it a unique and intricate appearance. The background of the image is a deep blue, emphasizing the vastness of the ocean and the creature's presence in its natural habitat. The overall style of the image is artistic and captivating, showcasing the beauty and complexity of the sea creature.

**Color caption with GPT-4 for GT input:**
A cool-toned underwater scene dominated by blues and browns. Jellyfish @ center: dark brown with pale white spots. Water @ background: muted blue-green. Object @ top-right: bright yellow.

Figure 6: Analysis of different captioning strategies on semantic guidance for SDXL.

The visual results in Fig. 7 present the effectiveness of using fine-tuned LLaVA captions as semantic guidance for UIE. In each example, the input underwater image shows heavy color distortion, while the ground truth provides the desired clean appearance. The captions produced by our fine-tuned LLaVA accurately capture localized color differences (e.g., "diver's wetsuit @ center: deeper black," "coral @ midground: pinkish hue," "water @ entire frame: less green, more natual tones"), which serve as domain-invariant semantic cues. When these captions are employed to guide SDXL for high-quality generation, the enhanced outputs (last column) closely align with the ground truth in both global color balance and localized corrections. Overall, the figure highlights that textual color

difference captions effectively bridge the gap between degraded underwater inputs and clean-domain reconstructions, ensuring both fidelity and controllability.

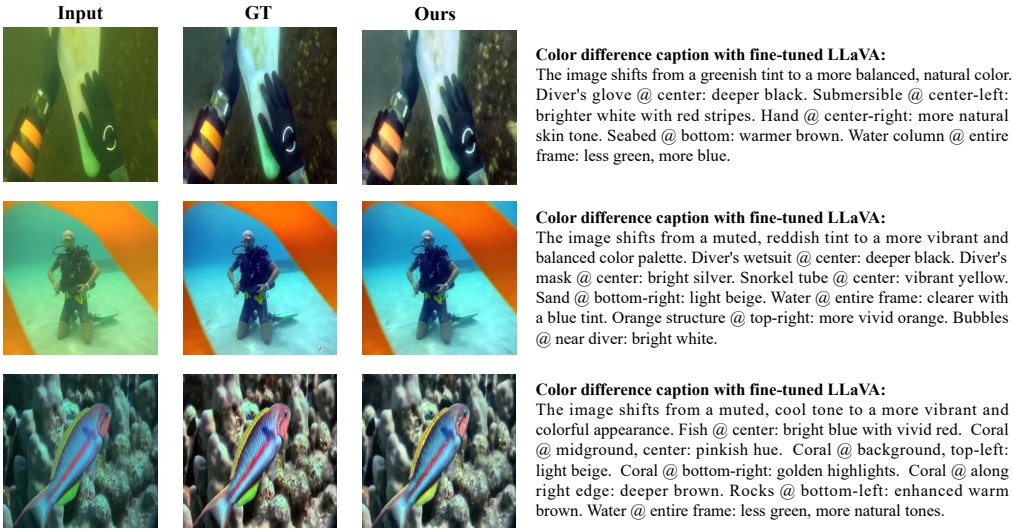

Figure 7: Visual examples with color difference caption generated by fine-tuned LLaVA on UIEB.

## A.2 VISUAL COMPARISONS FOR ABLATION STUDY

The visual results for ablation study in Fig. 8 shows that Case1 brings severe color deviation. Case2–3 reduce distortion yet introduce unnecessary artifacts. Case4 alleviates artifacts but appears unnatural. Case5 and Case6 enhance details and smoothness respectively, though with residual imbalance. Case7 achieves the best trade-off in color fidelity and structure preservation, producing results closest to the ground truth, thus confirming the effectiveness of the proposed model.

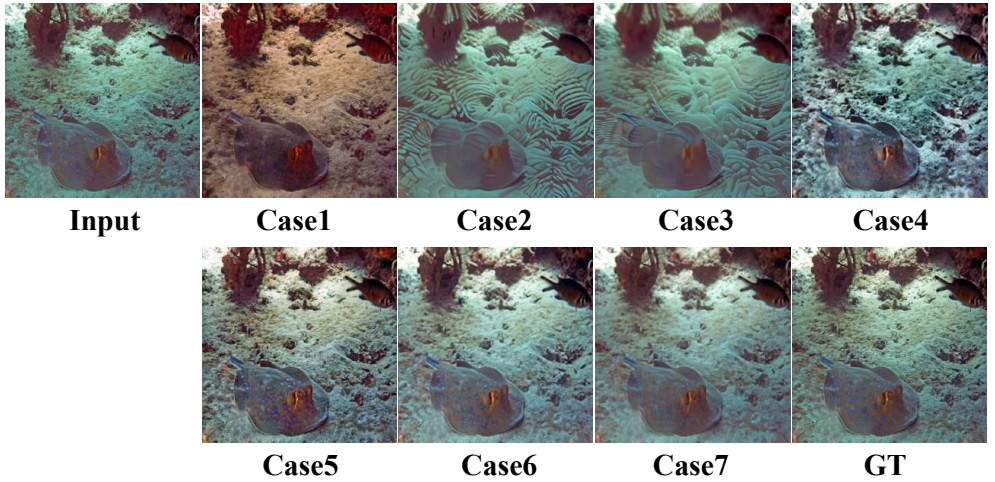

Figure 8: The visual comparisons for ablation study.

## A.3 ANALYSIS OF FEATURE MODULATION PARAMETERS

Here, we give a more detailed analysis of the learned $\gamma$ and $\beta$ in the adaptive degradation-aware feature modulation module. As illustrated in Fig. 9, we select 10 feature channels to present the corresponding value variation of $\gamma$ and $\beta$ during the denoising process. It is observed that the features of different channels can be scaled and shifted to varying degrees after transformation.

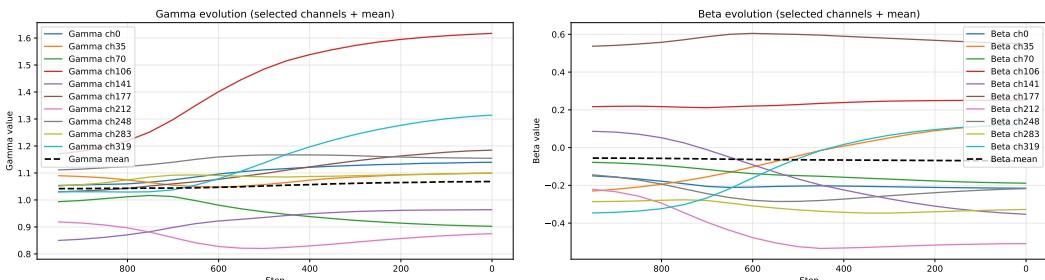

Figure 9: The distribution curves of $\gamma$ and $\beta$ during the denoising process. Zoom in for better view.

## A.4 QUALITATIVE RESULTS

In this subsection, we provide additional visual comparisons between our method and state-of-the-art methods on UIEB, EUVP, C60 and U45 datasets. As illustrated in Fig. 10, Fig. 11, Fig. 12 and Fig. 13, our results show more clear or comparable recovery performance compared to other SOTA methods both in color fidelity and the texture details.

## A.5 USE OF LLMS

Large Language Models (LLMs) were used to aid in the writing and polishing of the manuscript. Specifically, we used an LLM to assist in refining the language, improving readability, and ensuring clarity in various sections of the paper. The model helped with tasks such as sentence rephrasing, grammar checking, and enhancing the overall flow of the text.

It is important to note that the LLM was not involved in the ideation, research methodology, or experimental design. All research concepts, ideas, and analyses were developed and conducted by the authors. The contributions of the LLM were solely focused on improving the linguistic quality of the paper, with no involvement in the scientific content or data analysis.

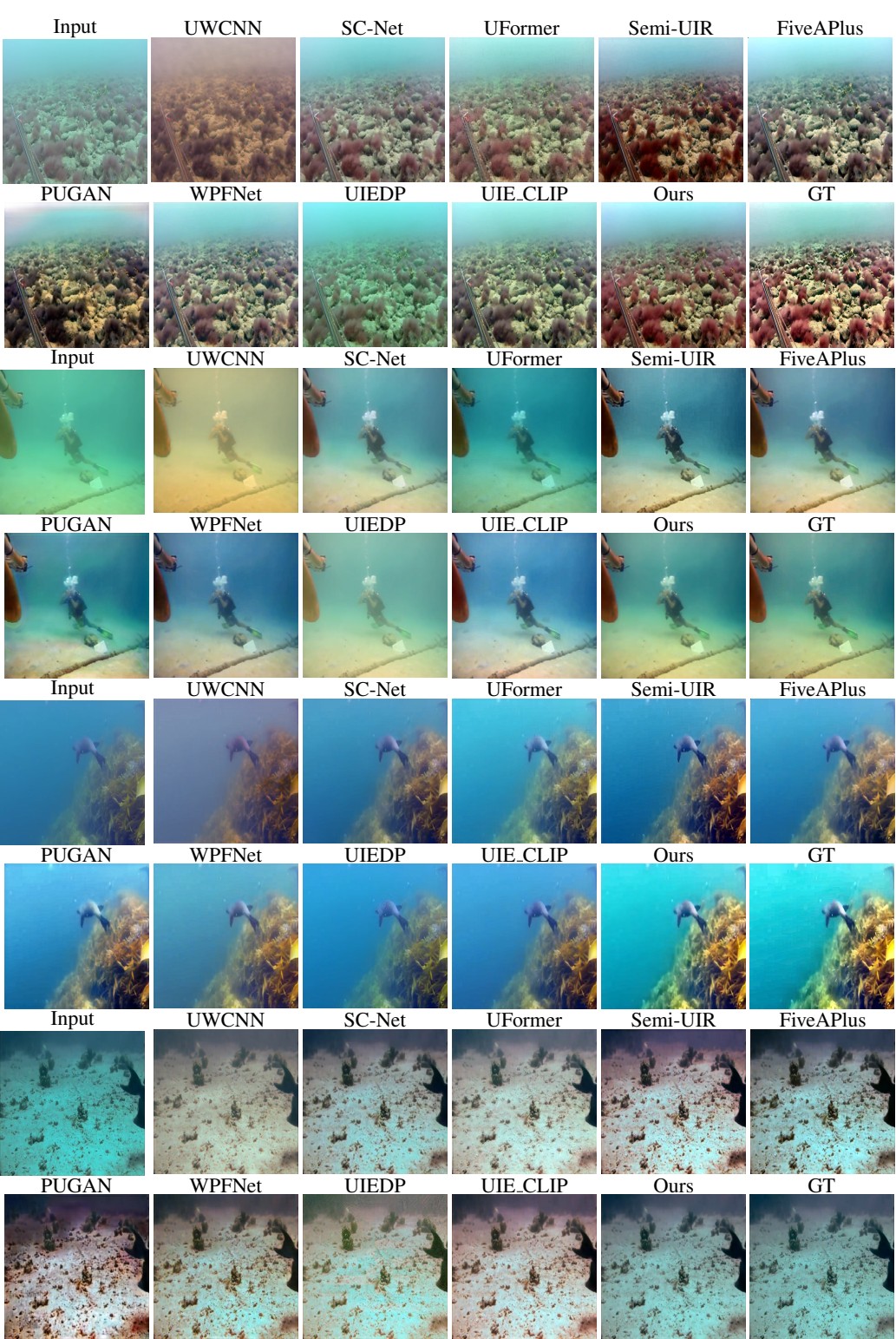

Figure 10: Visual comparisons of our CoDe with other state-of-the-art methods on UIEB dataset.

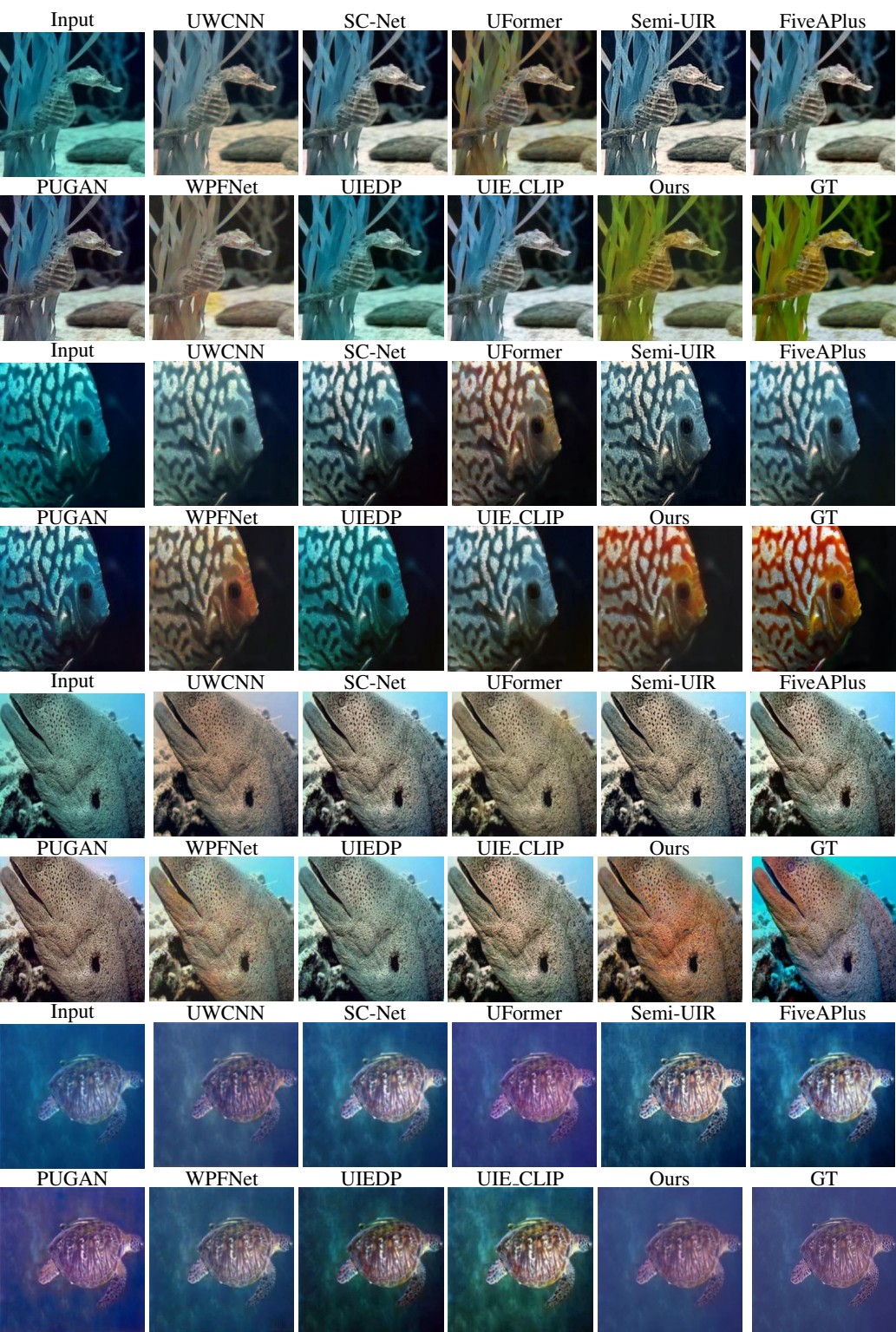

Figure 11: Visual comparisons of our CoDe with other state-of-the-art methods on EUVP dataset.

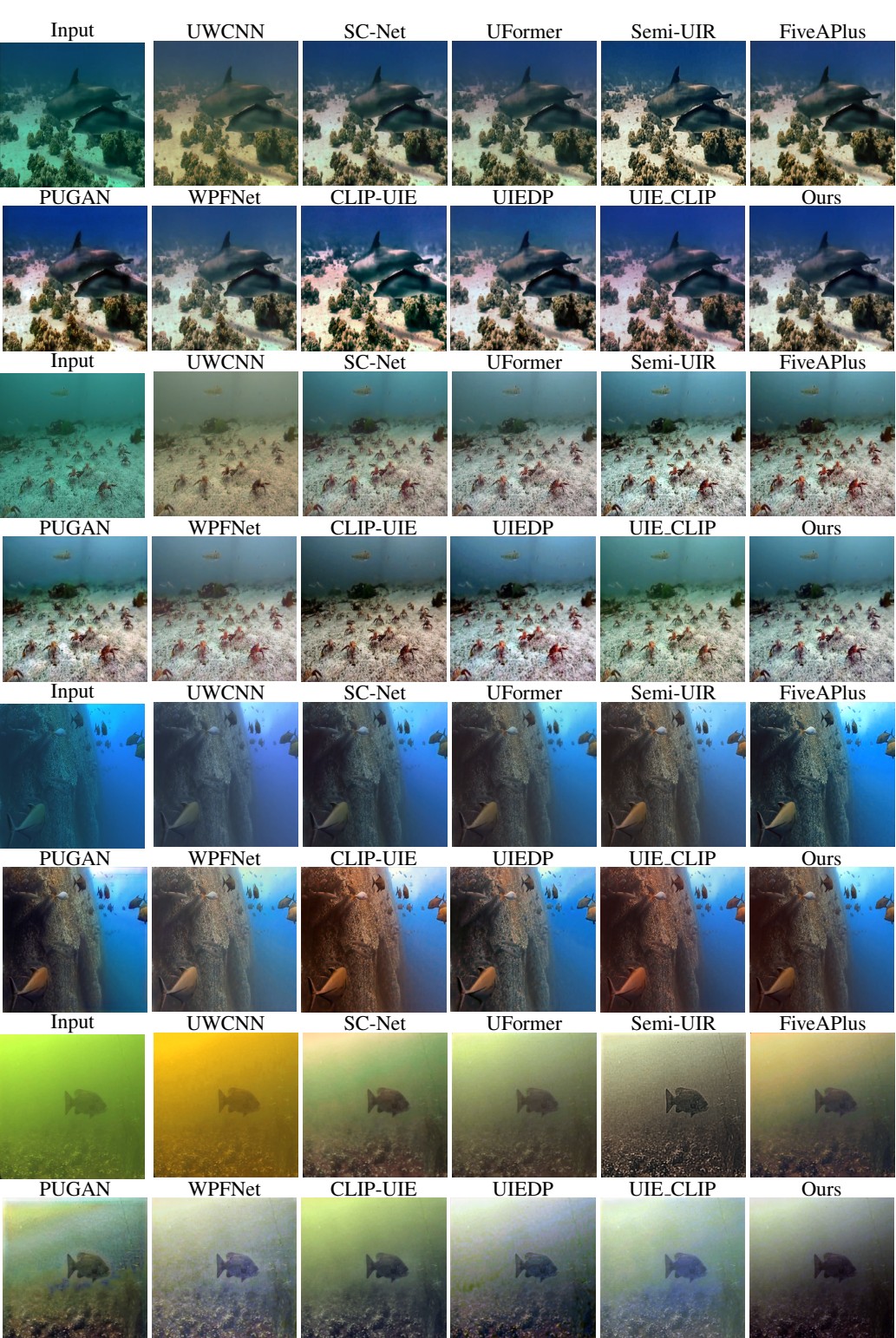

Figure 12: Visual comparisons of our CoDe with other state-of-the-art methods on C60 dataset.

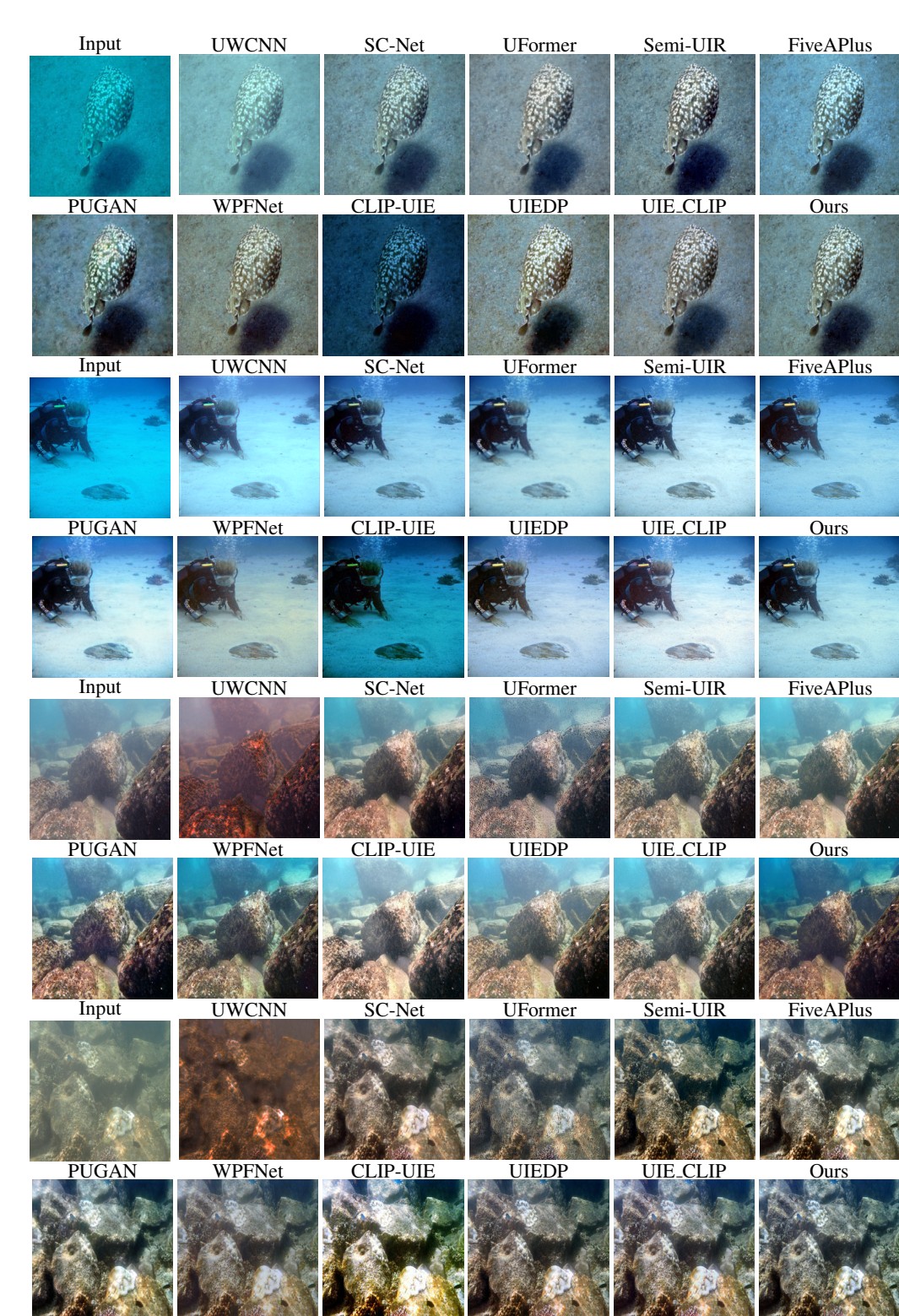

Figure 13: Visual comparisons of our CoDe with other state-of-the-art methods on U45 dataset.

