# OpenReview forum: "CoDe: Semantic Color Reasoning and High-Fidelity Detail Synthesis for Underwater Image Enhancement"
_ICLR.cc/2026/Conference — ICLR 2026 Conference Withdrawn Submission_

### Official Review · Reviewer_PeZr · 2025-10-26

**Soundness:** 3
**Presentation:** 3
**Contribution:** 3
**Rating:** 4
**Confidence:** 3

**Summary:**

This paper proposes a multi-modal decoupling framework for underwater image enhancement that disentangles color and texture restoration. The approach fine-tunes LLaVA to extract semantic color descriptions from degraded underwater images, uses these descriptions to guide SDXL for texture recovery, and introduces an ADFM module to fuse underwater features with diffusion latents. The method achieves SOTA results on multiple UIE benchmarks.

**Strengths:**

1. The paper clearly identifies that color correction and texture restoration are fundamentally different tasks in UIE, with color distortions requiring semantic-level reasoning while texture degradation needs fine-grained pixel-level modeling.

2. The pipeline elegantly connects color semantic extraction (fine-tuned LLaVA) → text-guided generation (SDXL) → adaptive feature fusion (ADFM), where each component addresses a specific aspect of the problem.

3. The evaluation spans both reference-based (PSNR/SSIM) and no-reference metrics (UIQM/UCIQE) across four underwater benchmarks, with consistent improvements and extensive ablations.

**Weaknesses:**

1. The method uses GPT-4 to generate color captions as supervision for LLaVA fine-tuning, but at inference time LLaVA must estimate these captions without access to clean references, creating a train-test mismatch. The paper lacks quantitative analysis of caption quality degradation and provides no evidence that fine-tuned LLaVA truly learns domain-invariant semantics rather than memorizing GPT-4's specific language patterns.

2. While the ADFM module design is structured, the paper provides weak justification for why this specific combination effectively suppresses noise artifacts during iterative denoising. Critically, the gating mechanism (Eq. 3) lacks explicit constraints to guarantee adaptive reduction of underwater feature influence as denoising progresses.

3. The gradual transition from clean captions to LLaVA captions during training (Section 3.4) lacks crucial implementation details—what is the transition schedule? How is the ratio configured?—yet this curriculum learning strategy is presented as essential without ablation or justification compared to direct LLaVA caption training.

4. This paper does not provide data on inference time, GPU memory consumption, or speed comparisons against CNN/GAN baselines. This makes it impossible to assess practical feasibility for time-critical applications like underwater robotics.

5. On UIEB, CoDe's PSNR gain over CLIP-UIE is only +0.22 dB (Table 1), and qualitative differences in Figure 3 are subtle.

**Questions:**

Please see weaknesses.

---

### Official Review · Reviewer_TEkn · 2025-10-30

**Soundness:** 2
**Presentation:** 3
**Contribution:** 3
**Rating:** 4
**Confidence:** 5

**Summary:**

This paper proposes CoDe, a multi-modal framework for underwater image enhancement (UIE), which leverages a vision-language model (LLaVA) to extract semantic color cues and a diffusion model (SDXL) for high-fidelity texture recovery. An Adaptive Degradation-aware Feature Modulation (ADFM) module is introduced to fuse underwater and clean features during denoising, effectively suppressing noise propagation. The authors carried on the experimental demonstration in the reference datasets and the non reference datasets, and has a great performance in the color fidelity and texture recovery.

**Strengths:**

1.	Novel decoupling of color and texture restoration: LLaVA is employed for semantic color reasoning and SDXL for texture synthesis, which is a innovative multi-modal strategy.

2.	Well-designed modulation module: ADFM adaptively fuses features without directly injecting degraded information, enhancing generation quality.

3.   Comprehensive experiments: The method is thoroughly evaluated on multiple datasets, showing consistent improvements in both full-reference and no-reference metrics.

**Weaknesses:**

1.  Effectiveness of color deviation captions: the method of LLaVA training relies on paired underwater clean images to generate color difference captions, which limits the generalization to unpaired scenes. Meanwhile, whether the color deviation captions can be quantified from the RGB three channels of image features, rather than using the GPT-4 model. The captions provided by GPT-4 model contains the color information and object information of the image, but the underwater image does not contain only one object. When there are too many objects, such as multiple fish and multiple objects, these situations are not fully discussed in the paper.

2.  Slow inference speed: Real time is necessary for underwater exploration, monitoring and other applications. Despite using LoRA for efficient fine-tuning, the SDXL-based diffusion model requires 20 sampling steps and 40000 iterations, and the two-stage model training hinders real-time applications.

3.  Sensitivity to LLaVA-generated captions: Inaccurate color descriptions from LLaVA may misguide the diffusion process and degrade enhancement quality. There is no quantitative discussion on the generated text description, which is relatively an ill-posed problem

**Questions:**

1.	The authors provided “the proposed framework can be readily extended to other crossdomain enhancement tasks such as low-light image enhancement and dehazing”, but without any experiments? How generalizable is the framework?
2.	The direct use of GPT-4 as the source of ground truth lacks justification. Other models like GPT-5, GPT-4 Plus, Deepseek, Gemini and so on.
3.	In the ADFM module, were other feature fusion mechanisms (e.g., cross-attention) explored? Why was the current combination (Channel Gating + Spatial Attention + FiLM) chosen? It remains unclear whether other feature fusion mechanisms (e.g., latest attention) were explored. The justification for the current combination of Channel Gating, Spatial Attention, and FiLM appears insufficient without ablations.
4.	The visual comparisons provided in the paper primarily showcase enhancements of single, isolated subjects. The lack of results on complex scenes (e.g., scenes with multiple coral species, fish schools, or intricate backgrounds) raises concerns about the model's robustness and generalization capability in more realistic and challenging underwater environments.
5.	As reflected in the loss function and the weaknesses mentioned earlier, the method still relies on paired images, making it difficult to demonstrate whether the disentanglement of color and texture actually contributes to the enhancement.

---

### Official Review · Reviewer_b4Ai · 2025-10-31

**Soundness:** 2
**Presentation:** 2
**Contribution:** 2
**Rating:** 2
**Confidence:** 5

**Summary:**

The paper proposes a multimodal underwater image enhancement approach by leveraging a fine-tuned LLaVA model for textual color description extraction and a diffusion-based generator for high-frequency and texture enhancement. The proposed method achieves high-quality image restoration on multiple benchmark datasets.

**Strengths:**

The paper presents a multimodal approach for underwater image enhancement, which utilizes a language model to extract textual information from images as supervisory guidance for image restoration.

**Weaknesses:**

The paper lacks a detailed introduction to the text extraction module (Figure 2). Relying solely on numerical experiments is insufficient to convince readers of the reliability and generalizability of the text color representations extracted by the LLaVA model.

**Questions:**

1. The paper lacks mathematical modeling of the textual feature representation. There is no verification of whether the textual information effectively describes the color characteristics of underwater images, nor how such validity should be evaluated.

2. What is the inference time of the diffusion probabilistic model-based texture enhancement method? The paper should provide comparative analysis with other approaches to demonstrate its computational efficiency advantages.

3. Underwater images lack ground truth; the so-called "ground truth" in Figure 3 cannot actually be used to evaluate the algorithm's effectiveness. From a visual perspective, the advantages of the proposed method are not significant.

4. The paper fails to provide essential explanations regarding "lightweight alignment": its specific meaning, implementation methodology, and evaluation criteria. This omission hinders comprehensive understanding and validation of the proposed approach.

---

### Official Review · Reviewer_GiWM · 2025-10-31

**Soundness:** 2
**Presentation:** 3
**Contribution:** 2
**Rating:** 2
**Confidence:** 5

**Summary:**

This paper proposed CoDe (Semantic Color Reasoning and High-Fidelity Detail Synthesis), an underwater image enhancement framework that decouples color correction and detail restoration to address color distortion and texture degradation. CoDe fine-tunes the LLaVA vision–language model to generate semantic color captions, guiding domain-invariant color correction, while the SDXL diffusion model restores high-frequency textures. An Adaptive Degradation-aware Feature Modulation (ADFM) module fuses underwater and latent features through attention and FiLM modulation to suppress artifacts, and LoRA enables efficient fine-tuning.

**Strengths:**

1. The paper introduced a novel color–texture decoupled underwater image enhancement framework that combined diffusion models with vision–language models, representing a creative and effective approach to handling complex underwater degradations.
2. By fine-tuning LLaVA to generate color-related textual descriptions, the method performed semantic color correction rather than relying solely on pixel-level cues, enabling domain-invariant and context-aware enhancement.
3. The integration of SDXL for texture synthesis ensured realistic and sharp detail restoration, outperforming traditional CNN- and GAN-based methods in perceptual quality.
4. The proposed ADFM module effectively fused underwater and latent diffusion features, using attention and FiLM modulation to suppress degradation artifacts and stabilize the denoising process.

**Weaknesses:**

1. The enhancement relied on LLaVA-generated color captions, which might be inaccurate or inconsistent under severe degradation, and the paper lacked evaluation of their reliability.
2. The study focused on visual quality but did not test whether or not enhancement improved downstream tasks, such as detection or segmentation, limiting its practical relevance.
3. Although LoRA reduced the training cost, SDXL inference remained computationally heavy and slow, restricting real-time or on-device deployment.
4. The network architecture figure lacked sufficient detail and were overly generalized; some key modules (e.g., FiLM and ResFuse) were not illustrated, making it difficult to understand the implementation process.

**Questions:**

1. It remains unclear how the fine-tuned LLaVA specifically focuses on color semantics while avoiding entanglement with texture or structural cues during caption generation. The paper should clarify what training constraints or supervision ensure this selective color reasoning.
2. The study only employed LLaVA for color caption generation without comparing it to other vision–language or large language models (e.g., BLIP-2, or GPT-4V). Such comparisons would better validate whether LLaVA is indeed the most suitable choice for semantic color extraction.
3. After introducing the color captions into the diffusion model, the paper did not provide attention maps or feature visualizations to illustrate how textual color guidance interacts with image features. Visual evidence would help confirm that the semantic cues effectively influence color correction and texture synthesis.

---

### Note · Authors · 2026-04-20

I have read and agree with the venue's withdrawal policy on behalf of myself and my co-authors.

---

### Meta-Review · Area_Chair_JfNV · 2025-12-13

**Summary:**

The rejection of this paper is based on collective critical concerns from four reviewers, which undermine the paper’s credibility and contribution:
1. The LLaVA-based semantic color caption module has a train-test mismatch (relying on GPT-4’s clean image captions for training but generating without references during inference) with no quantitative analysis of caption quality degradation. LLaVA’s suitability is unvalidated (no comparison with other vision-language models), and GPT-4 as the sole caption ground truth lacks justification.

2. The paper only reports visual quality metrics without evaluating downstream task performance. It lacks tests on complex underwater scenes (e.g., fish schools, multiple corals) and ablation experiments for key modules (e.g., ADFM’s feature fusion combination). The validity of claimed "ground truth" is also questionable.

3. The SDXL-based diffusion model suffers from slow inference speed (20 sampling steps, 40,000 iterations) and two-stage training, making real-time/on-device deployment unfeasible. No data on inference time, GPU memory, or comparisons with CNN/GAN baselines are provided.

4. Key modules (e.g., FiLM, ResFuse) are not illustrated in architecture figures; critical concepts (e.g., "lightweight alignment") and training strategy details (e.g., caption transition schedule) are missing. Mathematical modeling of textual feature representations is absent.

**Reviewer Concerns:**

Since the author did not provide a rebuttal to resolve these issues, all the concerns remain.

**Reviewer Scores:**

Since the author did not provide a rebuttal to resolve these issues, all the concerns remain.

---

### Decision · Program_Chairs · 2026-01-26

Reject